# Characterization and Preliminary In Vitro Antioxidant Activity of a New Multidrug Formulation Based on the Co-Encapsulation of Rutin and the α-Acylamino-β-Lactone NAAA Inhibitor URB894 within PLGA Nanoparticles

**DOI:** 10.3390/antiox12020305

**Published:** 2023-01-28

**Authors:** Agnese Gagliardi, Silvia Voci, Nicola Ambrosio, Massimo Fresta, Andrea Duranti, Donato Cosco

**Affiliations:** 1Department of Health Sciences, University “Magna Græcia” of Catanzaro, Campus Universitario “S. Venuta”, 88100 Catanzaro, Italy; 2Department of Biomolecular Sciences, University of Urbino Carlo Bo, Piazza del Rinascimento 6, 61029 Urbino, Italy

**Keywords:** antioxidant, NAAA inhibitors, nanoparticles, PLGA, rutin, URB894

## Abstract

A biodegradable and biocompatible polymeric matrix made up of poly(d,l-lactide-co-glycolide) (PLGA) was used for the simultaneous delivery of rutin and the (*S*)-*N*-(2-oxo-3-oxetanyl)biphenyl-4-carboxamide derivative (URB894). The goal was to exploit the well-known radical scavenging properties of rutin and the antioxidant features recently reported for the molecules belonging to the class of *N*-acylethanolamine-hydrolyzing acid amidase (NAAA) inhibitors, such as URB894. The use of the compounds, both as single agents or in association promoted the development of negatively-charged nanosystems characterized by a narrow size distribution and an average diameter of ~200 nm when 0.2–0.6 mg/mL of rutin or URB894 were used. The obtained multidrug carriers evidenced an entrapment efficiency of ~50% and 40% when 0.4 and 0.6 mg/mL of rutin and URB894 were associated during the sample preparation, respectively. The multidrug formulation evidenced an improved in vitro dose-dependent protective effect against H_2_O_2_-related oxidative stress with respect to that of the nanosystems containing the active compounds as a single agent, confirming the rationale of using the co-encapsulation approach to obtain a novel antioxidant nanomedicine.

## 1. Introduction

The development of nanoformulations able to simultaneously deliver two or more compounds within the same structure (combinational nanomedicine) has been amply researched in recent decades with the aim of maximizing the pharmacological efficacy of various drugs [1,2]. Usually, the choice of the compounds to be co-entrapped in a drug carrier involves the evaluation of their physicochemical properties and their potential interactions with the (bio)materials used for the delivery [3,4]. Various technological challenges can occur, as is true in the case of the polarity of compounds and their water solubility that influence (i) the choice of the solvent required for their solubilization during sample preparation, (ii) the type of procedure to be used for the development of the formulation and (iii) the nature of additional components and stabilizers of the nanosystems [1].

Investigations concerning the use of systems made up of synthetic copolymers for the protection and delivery of antioxidant compounds are relatively few as compared to other classes of bioactives, especially when considering the nanoformulations containing two or more drugs. The interest towards these classes of molecules has grown due to their various fields of application and biological relevance in different pathological conditions [5,6]. In fact, it has been reported that an oxidant/antioxidant imbalance can be detrimental for several biomolecules such as lipids, DNA and proteins and it is known to promote the development of many chronic diseases such as cancer, type 2-diabetes, cardiovascular conditions and Alzheimer‘s disease [7,8].

Generally, antioxidant compounds are classified as a function of different parameters, such as (i) their source (natural or synthetic, endogenous or exogenous), (ii) their molecular weight (high or low), (iii) their hydrophilicity/lipophilicity and (iv) according to their mechanism of action (enzymatic or non-enzymatic) [9]. Despite these differences, such molecules share a consistent degree of degradation during storage and, in the case of lipophilic compounds, poor aqueous solubility and scarce bioavailability following their administration in the free form, which are additional issues that further compromise their in vivo application. Nanotechnology has provided a solution to these problems, allowing the protection and modulation of the biological fate of the molecule(s) when they are entrapped within a drug delivery system.

In this context, the exploitation of poly(d,l-lactic-co-glycolic acid) (PLGA) for drug delivery purposes has been promoted by the peculiar and attractive features of this copolymer. Indeed, its favorable regulatory status (being approved for human use by the Food and Drug Administration and the European Medicines Agency), the safe degradation profile with two endogenous molecules (lactic and glycolic acids, respectively), combined with the possibility to select the ratio of the two monomers in order to modulate the desired technological/pharmacological outcomes, have favored the development of several commercially available PLGA-based products [10,11].

Among the natural-based antioxidant agents, rutin or vitamin P, a phytochemical derivative made up of quercetin as the aglycone moiety, glucose and rhamnose (Figure 1) and endowed with multiple pharmacological activities [12,13], is an example of a bioactive compound that can be entrapped within PLGA systems. In fact, several experimental works have described the encapsulation of this natural flavonoid within PLGA nanoparticles through different techniques, such as the nanoprecipitation method [14], the single- [15] or double- [16] emulsion technique, the microfluidic approach [17] as well as the double emulsion–evaporation method [18].

PLGA was also recently used by our own research team to develop nanoparticles containing the synthetic derivative URB866, an *N*-Acylethanolamine-hydrolyzing Acid Amidase (NAAA) inhibitor resulting in a formulation characterized by significant antioxidant activity [19]. NAAA is an N-terminal cysteine nucleophile hydrolase belonging to the choloylglycine hydrolase family able to degrade the peroxisome proliferator-activated receptor-α agonist palmitoylethanolamide (PEA), a molecule involved in the balance of physiological processes, such as those related to inflammation and pain [20]. α-Acylamino-β-lactone NAAA inhibitors are a promising class of molecules in the inflammation and analgesic field because of their ability to restore the levels of PEA after exposure to pro-inflammatory stimuli in several different cell lines [21,22]. Based on the need to confirm that the antioxidant properties of PLGA nanoparticles containing the *N*-(2-oxo-3-oxetanyl)amide NAAA inhibitor URB866 are to be considered typical of the whole class [19], a new molecule was tested in this investigation. The compound, namely (*S*)-*N*-(2-oxo-3-oxetanyl)biphenyl-4-carboxamide (URB894) [21], has the same core and similar characteristics with respect to the previous molecule, such as the ability to inhibit the enzymatic (IC_50_ = 115 ± 13 nM) and chemical features (Appendix A).

However, from a structural comparison it can be considered that, in the case of URB894, the second aromatic ring is not fused to the first one via a double bond but rather linked by a single bond to the second ring located in the para position (Appendix A). This variation is important because it induces different conformational properties and makes the drug more flexible, thus probably better able to adapt itself to the size of the hydrophobic binding pocket and then to influence the rate of inhibition of the enzyme (IC_50_ = 115 vs. 160 nM). The importance of steric requirements is confirmed by the fact that the *m*-biphenyl isomer of URB894 poorly (IC_50_ = 4400 ± 1200 nM) inhibits the activity of NAAA [23]. Moreover, the splitting of a benzene-based portion of the type described (called benzo-splitting) is a design strategy that could lead to the modification not only of the solubility but also the pharmacokinetic profile and, sometimes, the toxicity; therefore, it is very important for the discovery of new molecules of pharmaceutical interest. On the other hand, the lipophilicity (clogP: URB894 = 2.37 vs. URB866 = 1.66) should be considered because it could influence the entrapment of the molecule inside a polymeric system.

Based on these findings and considering the results previously described by our research team [19,24], in this work the PLGA nanoparticles were exploited for the co-delivery of two molecules, rutin and URB894, used as models of natural and synthetic antioxidant compounds, respectively (Figure 1). The choice of this association is based on the fact that the (i) well-known radical scavenging features of rutin could be improved by association to the NAAA inhibitor and (ii) the PLGA systems have already been demonstrated to be useful carriers for the multidrug delivery of various active compounds [25,26].

In detail, a preliminary physicochemical and technological characterization was performed with the aim of selecting the best concentration of URB894 to be used in association with rutin during the preparation of the polymeric nanosystems. Successively, the in vitro antioxidant features of the resulting multidrug formulation were assayed on two human normal cell lines using hydrogen peroxide as stressing agent.

## 2. Materials and Methods

### 2.1. Materials

PLGA (75:25, molecular weight 66,000–107,000 Da), acetone, 3-[4,5-dimethylthiazol-2-yl]-3,5-diphenyltetrazolium bromide salt (used for MTT tests), dimethylformamide (DMF), phosphate-buffered saline (PBS) tablets, dimethyl sulfoxide (DMSO), methanol (CH_3_OH), methylene chloride (CH_2_Cl_2_), petroleum ether, 4-phenylbenzoic acid, tetrahydrofuran (THF), triethylamine (Et_3_N) and amphotericin B solution (250 μg/mL) were all purchased from Sigma Aldrich (Milan, Italy). Oxalyl chloride [(CO)_2_Cl_2_] was purchased from Alpha Aesar (Ward Hill, MA, USA). Dry THF was distilled over sodium and benzophenone. Dry DMF, CH_2_Cl_2_, and Et_3_N were used as supplied. Petroleum ether refers to alkanes with a boiling point 40–60 °C. Poloxamer 188 was obtained from BASF (Ludwigshafen, Germany). Human keratinocyte (NCTC-2544) and chondrocyte (C-28) cells were provided by the Istituto Zooprofilattico Sperimentale della Lombardia e dell’Emilia Romagna. Dulbecco’s Modified Eagle Medium (DMEM) supplemented with glutamax I, penicillin/streptomycin solution, fetal bovine serum (FBS) and trypsin/ethylene diamine tetraacetic acid (EDTA) were all purchased from GIBCO (Life Technologies, Monza, Italy).

### 2.2. Synthesis and Characterization of URB894

The synthetic procedure (Figure 2) and the characterization of URB894 were already described by Solorzano et al. [23].

### 2.3. Preparation of PLGA Nanoparticles

The preparation procedure of the PLGA nanosystems was performed according to the nanoprecipitation technique, as previously described [27]. Namely, PLGA (0.6% *w*/*v*) was solubilized in 2 mL of acetone and added to 5 mL of MilliQ water containing poloxamer 188 (1% *w*/*v*); this suspension was homogenized (24k rpm, 1 min, Ultraturrax model T25 IKA^®^, Werke Gmbh & Co., Staufen, Germany), and then stirred on a magnetic plate for 12 h at room temperature, in order to favor the evaporation of the organic solvent [27]. The final concentration of PLGA in the formulation was 2.4 mg/mL.

The drug-loaded PLGA nanoparticles were obtained as described above, adding URB894 (0.2–0.6 mg/mL) or rutin (0.2–0.8 mg/mL) in the organic phase as single agents or in association (Table 1).

The nanosystems were purified by ultracentrifugation (i) at 90k × g for 1 h (4 °C) using an Optima TL apparatus (Beckman Coulter s.r.l., Milano, Italy) in order to evaluate their mean sizes, polydispersity index and surface charge as well as to quantify the amount of each drug that was encapsulated within the carriers, or (ii) by using Amicon^®^ Ultracentrifugal filters (cut-off 10 kDa, Merck, Darmstadt, Germany) at 4k rpm for 20 min before the in vitro experiments [19].

### 2.4. Physicochemical Characterization

The average diameter, size distribution and surface charge of the various systems were evaluated by the dynamic light scattering (DLS) technique using a Zetasizer NanoZS apparatus (Malvern Panalytical Ltd., Spectris plc, Malvern, UK) by diluting each sample at a 1:50 ratio in MilliQ water [19]. Transmission Electron Microscopy (TEM) was used to perform the morphological characterization of the nanosystems [24].

The stability of the samples was investigated as a function of time and temperature with a Turbiscan Lab^®^ Expert (Formulaction, Toulouse, France) and expressed as variation of the Turbiscan Stability Index; data were processed using a TurbiSoft 2.0, as previously reported [28].

### 2.5. Evaluation of the Drug Entrapment Efficiency (EE), Loading Capacity (LC) and Release Profiles

The amount of URB894 and rutin retained by the nanosystems was quantified by UV-vis spectrophotometry (Lambda 35, Perkin Elmer, Waltham, MA, USA), as previously described [19,29]. A total of 1 mL of each sample was centrifuged as described in Section 2.3, and the pellet obtained at the end of this process was dissolved in acetone; the organic solvent was then removed using a nitrogen flux. The pellet was incubated in DMSO (6 h) in order to promote the leakage of each entrapped drug. Finally, the solution was analyzed at the λ_max_ of 345 and 362 nm for URB894 and rutin, respectively, using an empty PLGA formulation as blank [19,30].

The EE was expressed as the ratio % between the amount of drug that became entrapped within the nanoparticles (D_e_) with respect to the concentration of compound that was used during their preparation (D_a_), as reported below:EE (%) = D_e_/D_a_ × 100(1)

The I_2_ assay was used in order to calculate the amount of poloxamer 188 integrated in the colloidal structure [19]. The LC of the carriers was then calculated as the ratio between the amount of encapsulated compound (as single agents or in association) and the total weight of the nanoparticles, and expressed as a percentage according to the following equation:LC (%) = Amount of entrapped compound/Total weight of nanoparticles × 100(2)

In addition, the release kinetics of URB894 and rutin from the nanocarriers were investigated by means of the dialysis technique under physiological (pH 7.4) and acid conditions (pH 5), respectively [19,30]. In detail, 1 mL of each sample were placed into a dialysis bag (cut-off of 12–14 kDa, Spectrum Laboratories Inc., Eindhoven, The Netherlands) and then immersed in 200 mL release medium kept at 37 ± 1 °C and under moderate agitation. A total of 200 mL of a PBS/ethanol mixture (80:20 *v*/*v*) was used as a release medium and sink conditions were preserved [19,29]. At fixed time points, 1 mL of the receptor fluid was collected and replaced by fresh.

The amount of URB894 and rutin released over time was calculated by UV-vis spectrophotometry at the λ_max_ reported at the beginning of this section, and using the equation reported below:Release (%) = Drug*_rel_*/Drug*_load_* × 100(3)
where Drug*_rel_* is the amount of released drug and Drug*_load_* is the amount of URB894 or rutin entrapped within the PLGA nanoparticles.

### 2.6. Cell Viability and Treatments 

C-28 and NCTC-2544 cells were cultured as already described [19,30]. Briefly, each cell line was grown in plastic culture dishes (100 mm × 20 mm) using a water-jacketed CO_2_ incubator (Thermo Scientific, Dreieich, Germany) at 37 °C (5% CO_2_) in a DMEM culture medium with glutamine, enriched with penicillin (100 UI/mL), streptomycin (100 µg/mL), amphotericin B (250 µg/mL) and FBS (10% *v*/*v*).

The antioxidant activity of each molecule (as free form or encapsulated within PLGA nanoparticles as single agents and in association) was evaluated by means of MTT testing and using hydrogen peroxide (H_2_O_2_) as stressing agent [19,30]. Namely, 7 × 10^3^ cells/0.2 mL were seeded in a 96-well culture plate, pretreated with 5, 10 and 25 µM of the various formulations for 24 h, and then incubated with H_2_O_2_ 800 µM for 1.5 h.

Cell viability was then evaluated by adding 20 µL of MTT salts (solubilized in PBS at a concentration of 5 mg/mL) to each well, incubating the plate for 3 h at 37 °C, and then solubilizing the formazan salts with a DMSO/EtOH (50/50) solution. The plate was analyzed at 540 nm with a reference wavelength of 690 nm using a microplate spectrophotometer (xMARK^™^-BIORAD, Bio-Rad Laboratories Inc., Hercules, CA, USA). Empty PLGA nanoparticles were used as a blank. The resulting data are the mean of three different experiments calculated using the following equation:Cell viability (%) = (Abs_T_/Abs_C_) × 100(4)
where: Abs_T_ is the absorbance of treated cells, Abs_C_ is the absorbance of control (untreated cells).

### 2.7. DPPH Antioxidant Activity

The antioxidant activity of PLGA nanosystems containing the active compounds (as single agents or in association) were evaluated by means of the DPPH assay (Sigma Aldrich, Milan, Italy) [31]. Tests were performed by adding 500 µL of samples containing various concentrations of the formulations to 500 µL of a 0.1 mM ethanolic solution of DPPH. After incubation (1.5 h) in the dark, the absorbance was measured at a wavelength of 517 nm. The inhibition activity (inhibition %) was calculated as:DPPH inhibition (%) = 1 − A_sample_/A_c_ × 100(5)
where A_sample_ is the absorbance of the sample and A_c_ is the absorbance of the control (DPPH solution). Empty nanosystems have been used as a blank.

### 2.8. Statistical Analysis

One-way ANOVA was used to perform the statistical analysis of the various experiments considering significant a *p* value < 0.05. Results are reported as mean ± standard deviation.

## 3. Results and Discussion

### 3.1. Physicochemical Characterization of PLGA-Based Multidrug Carriers

Over the last decade, our research team has encapsulated several active compounds within PLGA nanoparticles by means of the nanoprecipitation technique, obtaining various formulations for potential use in pharmaceutical applications [27,32,33]. Among the molecules entrapped within the polymeric matrix, rutin [24] and URB866 [19] were observed to be efficiently retained by the nanosystems and their antioxidant features were preserved. Considering the results, the first step of this investigation was focused on the development of a PLGA multidrug carrier containing rutin and the NAAA inhibitor URB894.

Despite the fact that URB866 and URB894 share similar chemical features, the encapsulation of the second derivative within the PLGA nanosystems requires a new physicochemical characterization in order to evaluate the best concentration to be used in association with rutin. For this reason, the influence of various amounts of URB894 on the average diameter, size distribution and surface charge of PLGA nanoparticles was investigated and the results reported in Table 1.

Namely, the addition of the synthetic drug promoted a progressive increase in the mean diameter and polydispersity index (PdI) of the nanosystems; in detail, a PdI of ~0.3 was obtained when a concentration of 0.6 mg/mL of URB894 was used, and an additional increase in this parameter was observed when higher amounts of the synthetic compound were added to the organic phase of the formulation (data not shown). The same trend was observed for rutin, even though in this case it was possible to obtain a monodispersed population with up to 0.6 mg/mL of the compound (Table 1); indeed, a PdI value of ≈0.4 was observed when an amount of rutin equal to 0.8 mg/mL was used and is in agreement with previous experimental investigations [15,24].

On the other hand, the surface charge of the nanosystems was not influenced by the encapsulation of the two compounds, showing negative values (~−30 mV) as a consequence of the presence of the free carboxylic residues of the copolymer (Table 1) [34,35].

Based on these preliminary results, 0.4 mg/mL of URB894 was selected as the best concentration of the synthetic compound to be used for the development of the multidrug formulation; therefore, this amount of URB894 was associated to various concentrations of rutin (0.2–0.6 mg/mL) during the preparation of the nanosystems. As reported in Table 1, the co-encapsulation of URB and rutin evidenced the formation of nanosystems with average diameters below 200 nm and a monomodal distribution, suggesting that the presence of both molecules did not perturb the assembly of the polymeric matrix (Table 1). These findings were confirmed by TEM analyses which evidenced that the encapsulation of the active compounds did not modify the spherical shape and smooth surfaces of PLGA nanoparticles (Appendix A).

Once again, the surface charge of the samples did not show any significant variation with respect to the empty formulation. 

Additional studies were performed with the aim of evaluating the time- and temperature-dependent stability of the samples; in detail, the Turbiscan Stability Index (TSI) profiles of formulations containing the active compound(s) were compared to those of the empty nanoparticles in order to evaluate the influence of the drug concentration on the occurrence of physical phenomena, such as sedimentation, creaming, or flocculation. The use of URB894 did not induce a great variation in the TSI profiles of the samples as compared to the empty formulation, and the same was true when the analysis was performed at body temperature, suggesting a suitable stability of these nanosystems and confirming the results obtained by DLS analysis (Figure 3 and Table 1).

A similar trend was obtained using rutin up to a drug concentration of 0.6 mg/mL at room temperature (Figure 3C); higher amounts of the flavonoid and an increase in the temperature promoted a consistent variation of the TSI profiles (Figure 3), confirming also in this case the data previously obtained by DLS (Table 1). It was interesting to note that the sample prepared with 0.4 mg/mL of URB894 together with various amounts of rutin were characterized by the best TSI profiles, probably as a consequence of the effect provided by the synthetic compound on the colloidal structure; indeed, the slopes of the TSIs were similar to those of the samples prepared with URB894 as a single agent, and this was true also at 37 °C (Figure 3). The use of higher concentrations of the synthetic derivative associated to rutin promoted a significant destabilization of the nanosystems with the appearance of macroaggregates along with sedimentation (data not shown).

Considering the obtained data, it is plausible to affirm that the co-encapsulation of URB894 and rutin in PLGA nanoparticles can be obtained when specific concentrations of the active compounds are used during the sample preparation.

### 3.2. Evaluation of the Entrapment Efficiency, Loading Capacity and Release Profiles

The next phase of the study was based on the evaluation of the retention rate of the molecules encapsulated within the PLGA nanoparticles as single agents or in association, with the aim of selecting the best formulation to be used for the in vitro tests on human cells. Namely, the formulation characterized by the best entrapment efficiency of URB894 was obtained using an initial drug concentration equal to 0.4 mg/mL (Figure 4); in fact, ~55% of the compound was integrated into the polymeric structure (~0.22 mg/mL), while higher amounts of the synthetic derivative promoted a decrease in this parameter. The evaluation of the LC% of the nanosystems confirmed this trend, evidencing a drug loading equal to 1.6% when 0.4 mg/mL of URB894 was used, and no significant improvements in the retention features of the nanoparticles were obtained when the sample prepared with 0.6 mg/mL of the compound was analyzed; indeed, in this case the obtained LC was close to 1.8% (Figure 4A).

These findings suggest that the maximum amount of URB894 that can be entrapped within the PLGA matrix had been reached, providing the rationale of the increased mean diameter of the colloidal systems prepared with higher amounts of URB894 (Figure 4A). The data concerning the EE and LC herein described are slightly different with respect to those previously discussed by us when the analog URB866 was entrapped within the PLGA nanoformulation [19]. Probably, the steric hindrance determined by the biphenyl moiety of URB894 exerted a certain influence on the localization of the drug in the colloidal polymeric structure as compared to what occurred when the naphthalene portion of URB866 was present (Appendix A).

Contrarily, the encapsulation profiles of rutin revealed a progressive increase in the amount of the active compound retained by the nanoparticles; in detail, an EE% of close to 50% and a LC of 2% were obtained when 0.6 mg/mL of the flavonoid were used during the preparation of the nanosystems (Figure 4).

It was interesting to observe the trend that appeared when the quantification analyses were performed on the multidrug formulation made up of a fixed concentration of URB894 (0.4 mg/mL) and increasing amounts of rutin (0.2–0.6 mg/mL) (Figure 4B). 

The simultaneous presence of both compounds within the same colloidal structure evidenced a slight decrease in the amounts of URB894 that became encapsulated with respect to that obtained when the inhibitor was added as a single agent during the sample preparation, while the amount of rutin retained by the polymeric structure was quite similar to that previously described (Figure 4). Taking into consideration the different molecular weights of each compound (267.28 g/mol vs. 610.52 g/mol for URB894 and rutin, respectively), a greater number of URB894 molecules can be hosted within the matrix with respect to those of rutin, confirming the better affinity of the NAAA inhibitor for the colloidal structure. The data herein discussed, demonstrate that the association of 0.4 mg/mL of URB894 and 0.6 mg/mL of rutin during the preparation of the PLGA nanoparticles promoted the development of a colloidal formulation characterized by the best EE% (~50% and 40%, respectively) and LC values ~1.5% and 1.8%, respectively (Figure 4B).

It is interesting to note that the multidrug formulation was characterized by a molar concentration of URB894 that was two-fold higher than that of rutin as a consequence of their molecular weights.

Moreover, the leakage kinetics of URB894 and rutin, encapsulated as single agents or in association within PLGA nanosystems, were evaluated under physiological and acidic conditions (Figure 5); in particular, in view of the results previously discussed, the test was performed on the multidrug formulation prepared with 0.4 mg/mL of URB894 and 0.6 mg/mL of rutin.

The samples containing URB894 as a single agent confirmed a biphasic release pattern of the compound; in particular, the leakage of URB894 was characterized by an initial burst effect followed by a prolonged and sustained release from the PLGA matrix over time [19]. In detail, in the first 8h of analysis ~30–40% of the drug was found in the release media under physiological conditions when 200–600 µg/mL of URB894 were initially used (Figure 5). On the contrary, the use of an acidic environment promoted an increase of 10–15% of the cumulative amount of URB894 that was released over time (Figure 5).

The evaluation of the rutin-loaded samples evidenced a peculiar biphasic release pattern in this case too, a trend in good agreement with other PLGA-based formulations containing the natural compound that have already been described [16,17,18]. However, at pH 5, 60–70% of rutin was released during the first few hours (Figure 5). Contrarily, the release of rutin at pH 7.4 was significantly lower with respect to that previously discussed. These different trends can be explained as a consequence of the solubility of rutin at different pH conditions [34,35,36,37].

In the case of the multidrug formulation, the release profiles of the two active compounds were quite similar to those previously described; indeed, the biphasic release of the molecules was maintained even after their co-encapsulation, while the acidic environment promotes a higher drug leakage. The results confirm the possibility of obtaining a prolonged drug leakage of the entrapped drugs using PLGA nanoparticles and demonstrate the potential opportunity of exploiting the acidic pH to promote a higher localization of the active compounds into the inflamed area (Figure 5).

In view of these results, the systems prepared with 0.4 mg/mL of URB894 and 0.6 mg/mL of rutin, as single agents or in association, were chosen to be tested on human cells in order to preliminarily evaluate their antioxidant activity.

### 3.3. Protection of Human Cells from Hydrogen Peroxide-Induced Oxidative Stress and Antioxidant Activity

The physicochemical characterization of the nanosystems was propaedeutic to testing their protective features on two human cell lines, i.e., chondrocytes and keratinocytes, which could be potentially involved in oxidative stress reactions.

As can be observed in Figure 6, when URB894 and rutin were entrapped within PLGA nanoparticles as single agents, an increased dose-dependent protective effect against H_2_O_2_-related oxidative stress was observed. Indeed, cell viability between 50% and 60% was obtained when the two drugs were used in the free form in a concentration range of 5–25 μM, while an increase in this parameter was observed when they were encapsulated within the PLGA matrix (cell viability of 60–80%) (Figure 6). In addition, it was shown that the presence of both molecules within the same colloidal carrier promoted an additional improvement in cell protection, especially at the highest drug concentration (URB894 25 μM/rutin 13 μM). Indeed, this formulation gave a cell viability of ~90% and ~80% for NCTC-2544 and C-28 cells, respectively, demonstrating a strong protective activity against the induced oxidative stress (Figure 6). The empty formulation evidenced the lack of any protective features, as already described in previous experimental investigations [19,34].

This data is in good agreement with our recent findings [19] and can be related to the ability of the PLGA nanosystems to (i) enhance the cell uptake of their cargo molecules, mainly through fluid phase pinocytosis or clathrin-mediated endocytosis [37,38,39,40], and (ii) to promote a sustained release of the encapsulated molecules. Moreover, it should be mentioned that the entrapment of the active compounds within the colloidal carriers can preserve them from an early enzymatic degradation, allowing them to exert their antioxidant activity once the intracellular environment has been reached [34].

The evaluation of the antioxidant properties of URB894 and rutin encapsulated within PLGA nanoparticles was also performed by means of DPPH assay (Figure 7). The results confirmed that the encapsulation of URB894 and rutin did not compromise their radical scavenging activities. In particular, the use of 50 µg/mL of URB894 or rutin showed a percentage of DPPH inhibition of 55% and 65%, respectively, when they were encapsulated as single agents (Figure 7). The multidrug formulation evidenced a slight increase in this parameter (70%), confirming the rationale of using their association to promote a better pharmacological outcome.

## 4. Conclusions

The criticism related to the solubility and physical stability of α-acylamino-β-lactone NAAA inhibitors and rutin are important drawbacks that need to be addressed in order to be able to exploit the pharmacological activity of these molecules. Moreover, α-acylamino-β-lactone NAAA inhibitors evidenced low chemical and plasmatic stability [40], and despite the fact that various attempts have been made trying to improve these crucial aspects, no formulations are currently under investigation in clinical trials. In particular, URB894 was very promising because it not only showed an in vitro positive inhibitory capacity towards NAAA, but it also tended to reduce in vivo carrageenan-induced leukocyte infiltration [23]. Based on the examined scenario, interventions carried out with appropriate formulations could lead to an improvement in the properties and a greater use of URB894 and its chemical class. In this work, the approach of nanoencapsulation within PLGA nanoparticles, recently proposed for the analog molecule URB866, allowed us to obtain a novel multidrug formulation characterized by great antioxidant activity. In fact, the association of rutin in the same colloidal structure enhanced the antioxidant features of URB894, demonstrating the suitability of this formulation to exert a significant protective effect on cells stressed by hydrogen peroxide.

Considering the strict relationship between ROS and inflammation [41,42,43], these results are encouraging in light of the potential in vivo application of the proposed formulation for the treatment of acute inflammation. Indeed, several reports have already described the in vivo anti-inflammatory properties of rutin [30,44,45,46], which, combined with those of URB894 [23], could be used to obtain an innovative formulation to be used in nanomedicine for the treatment of inflammatory conditions. To reach this goal additional investigations are in progress in order to better understand (i) the specific interaction between the two active compounds and PLGA, (ii) the influence of their concentration in the polymeric matrix, and (iii) the pharmacokinetic profiles of the molecules contained in the multidrug formulation after systemic administration.

## Figures and Tables

**Figure 1 antioxidants-12-00305-f001:**
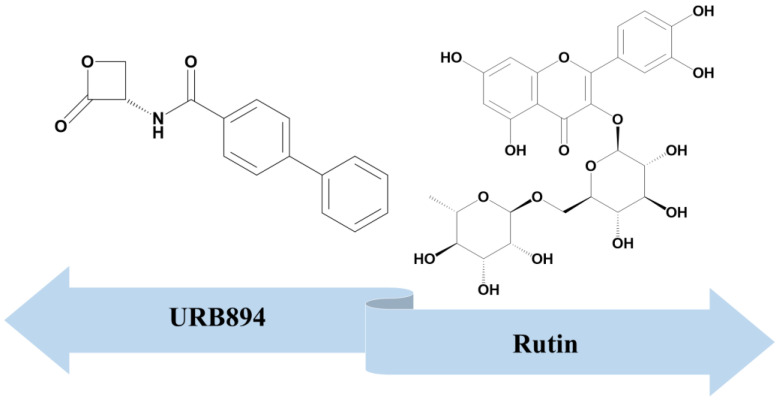
Chemical structures of the compounds co-encapsulated within PLGA nanoparticles.

**Figure 2 antioxidants-12-00305-f002:**
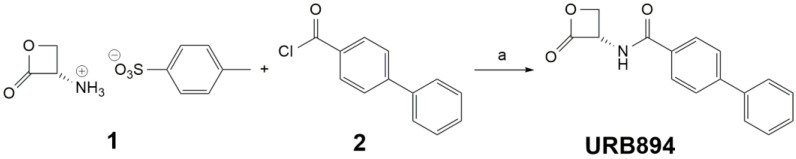
Reagents and conditions: (a) Et_3_N, THF, 0 °C, 0.5 h, then rt, 3 h.

**Figure 3 antioxidants-12-00305-f003:**
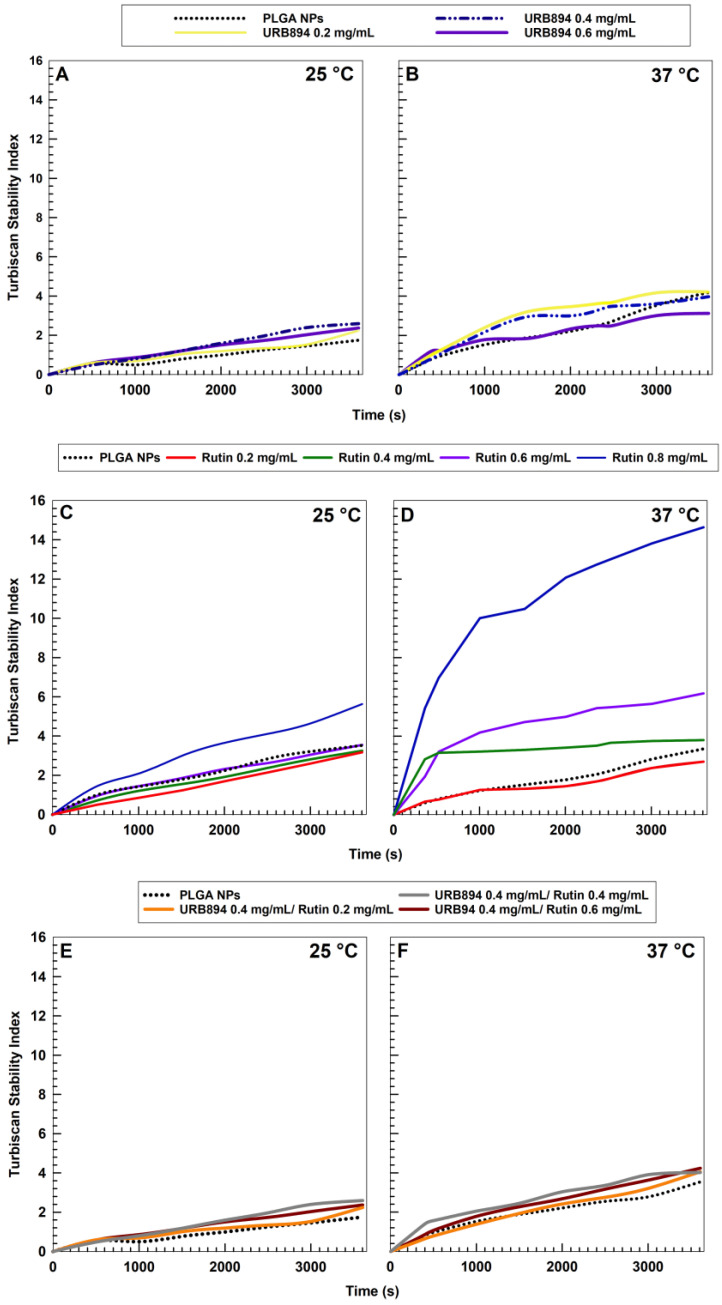
Turbiscan Stability Index (TSI) profiles of PLGA nanoparticles (NPs) prepared with various amounts of URB894 (0.2–0.6 mg/mL) or/and rutin (0.2–0.8 mg/mL) as single agents or as multidrug carriers (0.4 mg/mL of URB894 and 0.2–0.6 mg/mL of rutin, respectively) as a function of time and temperature. (**Panels** (**A**,**C**,**E**)): 25 °C. (**Panels** (**B**,**D**,**F**)): 37 °C. The reported result is a representative experiment of three independent tests.

**Figure 4 antioxidants-12-00305-f004:**
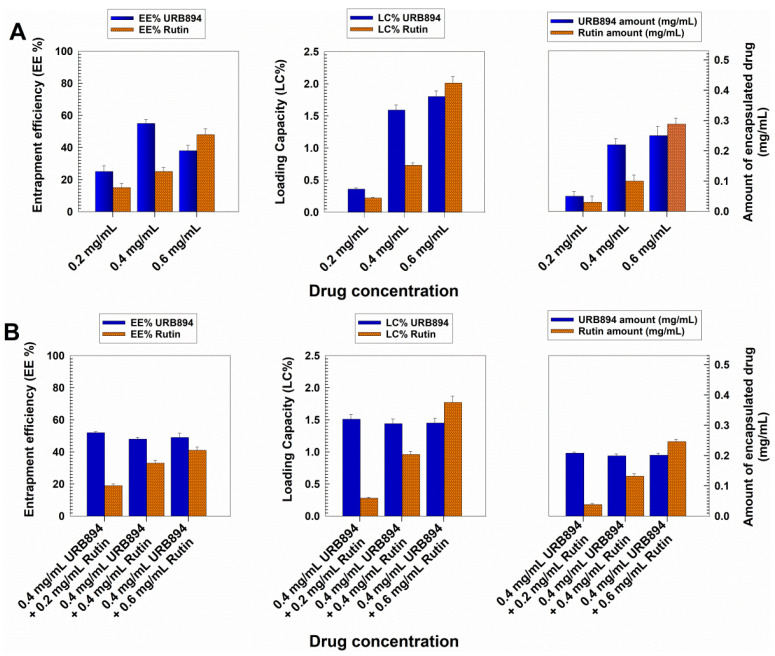
Entrapment efficiency and loading capacity of URB894 and rutin within PLGA nanoparticles encapsulated as (**A**) single agents or (**B**) in association in multidrug formulations as a function of the drug concentration used. Results are reported as the mean of three different experiments ± standard deviation.

**Figure 5 antioxidants-12-00305-f005:**
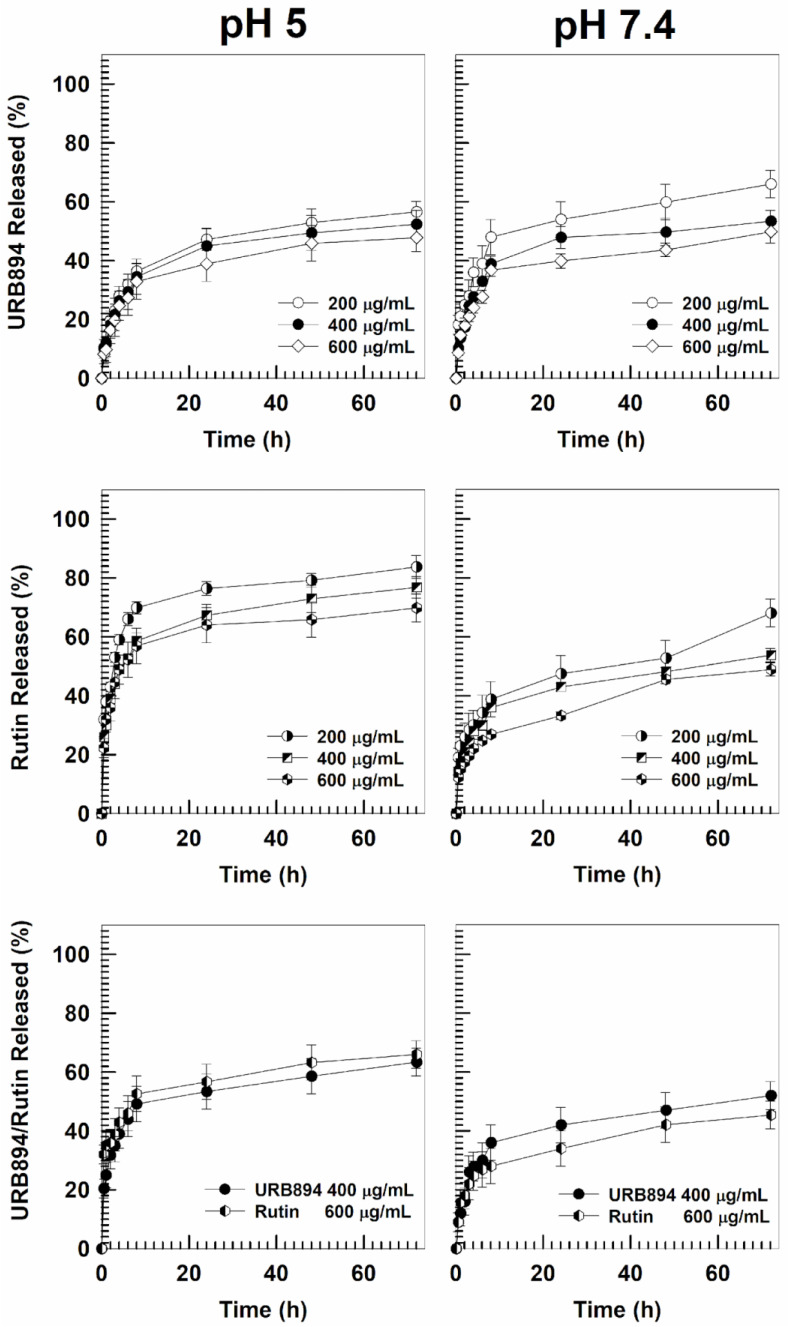
Release profiles of URB894 and rutin contained in PLGA nanoparticles as single agents or in association, in physiological and acidic conditions as a function of the amount of drug initially used and incubation time. The reported values represent the mean of three different experiments ± standard deviation. Error bars, if not shown, are within symbols.

**Figure 6 antioxidants-12-00305-f006:**
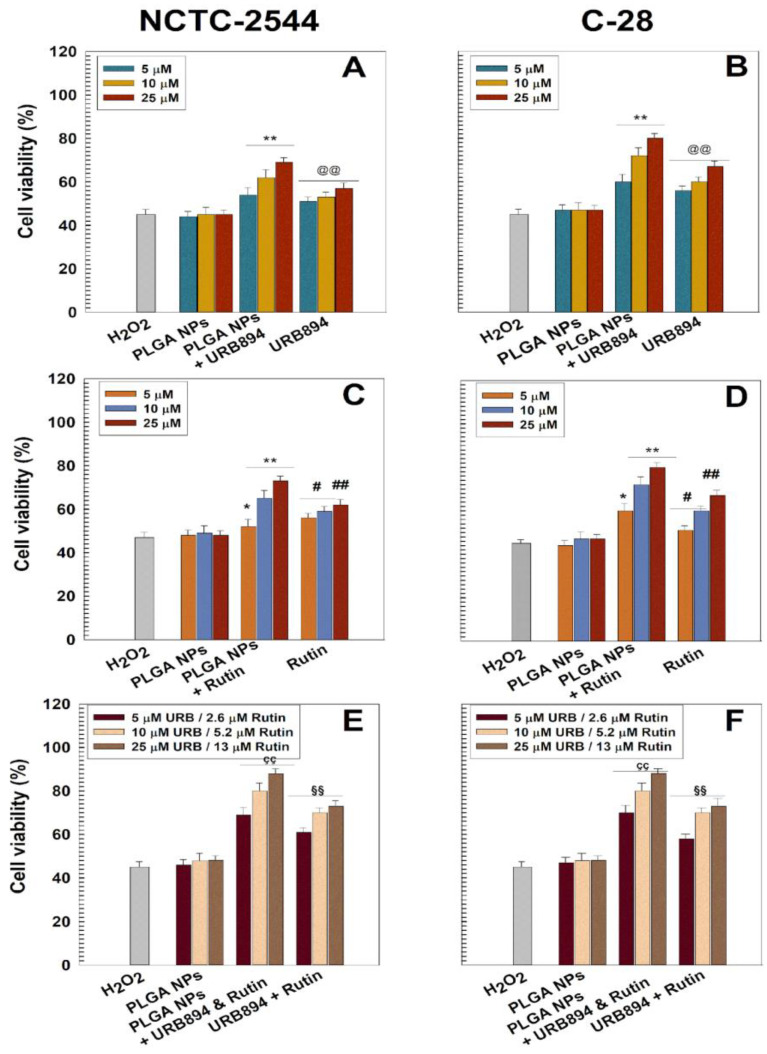
Cell viability during hydrogen peroxide-induced oxidative stress of URB894 (**Panels** (**A**,**B**)) and rutin (**Panels** (**C**,**D**)) as free form or encapsulated within PLGA nanoparticles (NPs) as single agents or as multidrug formulations (**Panels** (**E**,**F**)), evaluated by MTT testing on NCTC-2544 and C-28 cells as a function of the drug concentration. Results are reported as the mean of three different experiments ± standard deviation. The cells were incubated with different concentrations of drug(s) for 24 h and then treated with H_2_O_2_ (800 μM) for 1.5 h. *^,#,ç^ *p* < 0.05 and **^,##,çç,§§,@@^ *p* < 0.001: vs.·H_2_O_2_ 800 μM.

**Figure 7 antioxidants-12-00305-f007:**
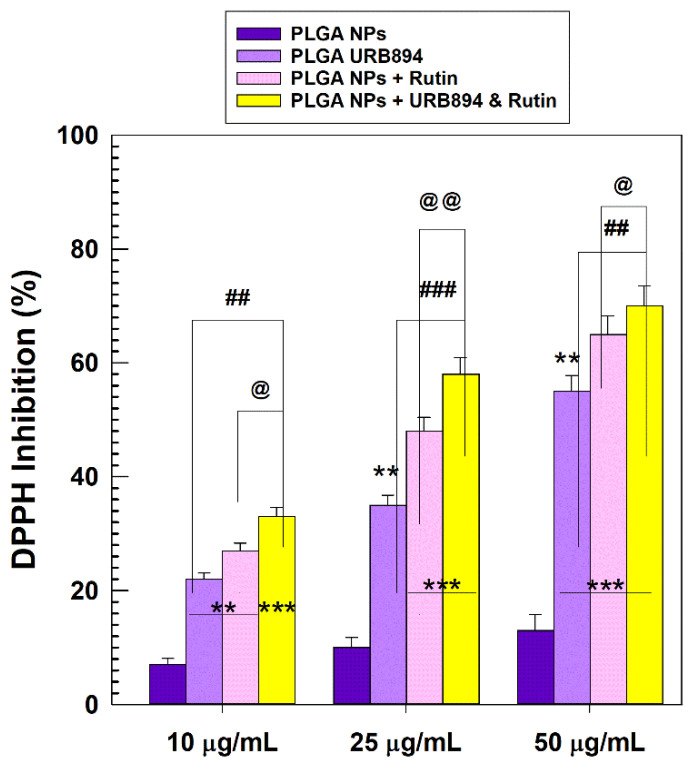
Antioxidant features of PLGA nanoparticles encapsulating URB894 (400 µg/mL) and rutin (600 µg/mL) as single agents or as a multidrug formulation evaluated by the DPPH assay. ** *p* < 0.001 and *** *p* < 0.0001 vs. empty PLGA nanoparticles. ^@^ *p* < 0.05 and ^@@^ *p* < 0.001 PLGA NPs + URB894 and Rutin vs. PLGA NPs + Rutin; ^##^ *p* < 0.01 and ^###^ *p* < 0.0001 PLGA NPs + URB894 and Rutin vs. PLGA NPs + URB894. Results are the average of three independent experiments ± standard deviation. The concentrations of the active compounds tested as a multidrug formulation are the following: 50 µg/mL URB894 and 45.76 µg/mL Rutin; 25 µg/mL URB894 and 22.88 µg/mL Rutin; 10 µg/mL URB894 and 9.15 µg/mL Rutin.

**Table 1 antioxidants-12-00305-t001:** Physicochemical properties of PLGA nanoparticles as a function of the amounts of active compound(s) used during their preparation.

URB894 (mg/mL)	Rutin (mg/mL)	Mean Sizes (nm)	Polydispersity Index	Zeta Potential (mV)
-	-	110 ± 2	0.06 ± 0.01	−32 ± 1
0.2	-	134 ± 2 *	0.19 ± 0.02 *	−34 ± 1
0.4	-	170 ± 1 **	0.25 ± 0.03 **	−32 ± 2
0.6	-	196 ± 3 **	0.32 ± 0.05 **	−33 ± 2
-	0.2	115 ± 6	0.14 ± 0.01 **	−32 ± 1
-	0.4	132 ± 6 *	0.18 ± 0.08	−28 ± 2
-	0.6	141 ± 5 *	0.24 ± 0.04	−31 ± 2
-	0.8	314 ± 4 **	0.36 ± 0.01	−29 ±3
0.4	0.2	119 ± 2	0.19 ± 0.03	−31 ± 2
0.4	0.4	142 ± 1	0.18 ± 0.05	−27 ± 1
0.4	0.6	150 ± 2 *	0.19± 0.07 *	−29 ± 2

* *p* < 0.05 and ** *p* < 0.001 vs. empty PLGA nanoparticles. Results are expressed as the mean of three independent measurements performed in triplicate on three different samples ± standard deviation.

## Data Availability

All data available are reported in the article.

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
