# Peer review of "Characterization and Preliminary In Vitro Antioxidant Activity of a New Multidrug Formulation Based on the Co-Encapsulation of Rutin and the α-Acylamino-β-Lactone NAAA Inhibitor URB894 within PLGA Nanoparticles"

_antioxidants, 2023, doi:10.3390/antiox12020305_

Round 1

Reviewer 1 Report

The article “Characterization and preliminary in vitro antioxidant activity of a new multidrug formulation based on the co-encapsulation of rutin and the α-Acylamino-β-lactone NAAA inhibitor URB894 within PLGA nanoparticles” by A. Gagliardi and colleagues reports the encapsulation of rutin and URB894 into PLGA NPs and their physico-chemical and in vitro characterization.

The article represents the prosecution of a previously published work with a similar drug (URB866, ref. 19). However, there is no real innovation from the nanotechnological point of view (just the encapsulation of an additional drug, rutin) to the already described URB894 (C. Solorzano et al., ref. 23). In addition, the characterization is quite poor. There is no drug release profile (at acidic and physiological pH), TEM images, and other physico-chemical characterizations that would allow a better understanding of the results.

Similarly, the cellular characterization is also quite poor (just an MTT test on NCTC-2544 and C-28 cells with subsequent addition of H2O2). Additional tests to evaluate the oxidative stress must be included to characterize more in detail the antioxidant activity of the drug-loaded NPs.

Furthermore, the hydrogen peroxide-induced oxidative stress is usually performed by first inducing the stress (the H2O2), and then evaluate the antioxidant effect after several time points (e.g. 6, 24, 48 h) and interpreted as a function of the drug release profile (missing in the characterization).

In order to quantitatively determine an additive or a synergistic effect, a mathematical model must be applied (e.g. https://doi.org/10.1177/1947601912440575, https://doi.org/10.1038/s41598-018-23321-6, etc).

The images are quite small and often difficult to read and interpret, they should cover the full width of the page. Also, in Figure 4 there is no need to report again the chemical structures of URB894 and rutin (they are already in Figure 1), it is better to enlarge the graphs to make them more readable.

Several editing errors (e.g. italics in chemical names) are present and must be corrected.

Reviewer 2 Report

Data presented in Table 1 (figure 4,5) result from the average of how many samples? Add to the caption table this information (e.g.: mean±SD, n=?).

The methods described in section 2.2. Synthesis and characterization of URB894, can be removed since it was already published and did not add any new to the work.

Please specify in section 2.4. Physico-chemical characterization, if the analysis was done after purification?

The characterization of nanoparticles needs to be more complete, namely the crystalline state of the drugs when encapsulated and the release kinetics.

The %EE is a parameter that depends on the initial amount of the drug added to the system, while the %DL  (% drug loading) (% of amount of drug respective to total amount of NPs)  that better characterize the system, please add it to the results.

In section 2.6. please add the source of the two cell lines C-28 and NCTC-2544 cells.

The method described for antioxidant effect in section 2.6 is a cell viability assay and not an antioxidant one. Furthermore, no assay of drug activity when encapsulated is presented.

Round 2

Reviewer 1 Report

The authors addressed many of the issues raised and performed additional experiments as requested (TEM analysis, release profile at physiological and acidic pH, and DPPH assay). Nevertheless, despite the improved scientific soundness of the article, some points still need to be clarified.

Improvement of the in vitro characterization was addressed by performing a DPPH assay with the NPs encapsulating URB894 and rutin. In this case the control assay with the same amount of free drug/s was not reported and must be included. Furthermore, did the authors observe a statistically significant improvement in the antioxidant effect by encapsulating the 2 drugs? As far as I understand, the statistical analysis was performed only against the empty PLGA NPs but not against the encapsulated single drugs (nor with the free single and simultaneously delivered drugs).

Reviewer 2 Report

The authors consider all observations made to the manuscript improving it. Thus, in my opinion, the article may be considered for publication.

Author Response

The Authors are very grateful to the Reviewer for the positive evaluation.